# Effective coupling of rapid freeze-quench to high-frequency electron paramagnetic resonance

**E. Gabriele Panarelli, Harmen van der Meer, Peter Gast, Edgar J J. Groenen** *

Department of Physics, Huygens-Kamerlingh Onnes Laboratory, Leiden University, Leiden, The Netherlands

* groenen@physics.leidenuniv.nl

**Data Availability Statement:** All data available on Figshare: 10.6084/m9.figshare.8982347.

**Funding:** The authors received no specific funding for this work.

## Abstract

We report an easy, efficient and reproducible way to prepare Rapid-Freeze-Quench samples in sub-millimeter capillaries and load these into the probe head of a 275 GHz Electron Paramagnetic Resonance spectrometer. Kinetic data obtained for the binding reaction of azide to myoglobin demonstrate the feasibility of the method for high-frequency EPR. Experiments on the same samples at 9.5 GHz show that only a single series of Rapid-Freeze-Quench samples is required for studies at multiple microwave frequencies.

## 1 Introduction

Determination of reaction rates and detection of short-lived intermediates of fast chemical reactions are an important goal in those fields that involve molecular chemistry. Kinetics and intermediates shed light on the mechanism of a reaction, which in turn may yield broader information about the chemical system under study. One possible stratagem to investigate chemical kinetics is to let the reaction unfold for controlled time steps and then "freezing" it. In this way it is possible to follow the decay and growth of reactants and products, or the evolution of reaction intermediates. One of the most widely used techniques to attain this is called Rapid Freeze-Quench (RFQ), in use since 1961 [1], which is often coupled to Electron Paramagnetic Resonance (EPR) in view of the paramagnetic nature of the intermediates of a great deal of chemical reactions.

A multi-frequency approach in EPR is of particular interest, namely when low-frequency experiments (e.g. those at the standard frequency of 9.5 GHz, called X-band) are combined with high-frequency ones (HF-EPR, e.g. those at microwave frequencies of 95 and 275 GHz). Such approach offers a better and more complete characterization of the paramagnetic system. However, collection of RFQ samples is–to say the least–problematic for applications in HF-EPR, because the size of HF resonant cavities is hugely reduced as compared to the standard 9.5 GHz EPR, thus making the sample holders and the sample volume dramatically small. For example, from 9.5 to 275 GHz the sample volume downsizes from 100 μL to 20 nL, i.e., by about $10^4$ times. It is therefore vital to develop a sample packing technique that guarantees an efficient, homogeneous, and reproducible sample collection in the small capillaries used as sample holders for HF-EPR.

**Competing interests:** The authors have declared that no competing interests exist.

Schünemann *et al.* [2] are the first to report the application of RFQ to HF-EPR in a multi-frequency EPR study (at 9.6, 94, 190, and 285 GHz). The method used by the authors to pack the RFQ samples for HF-EPR is basically the same as for low frequency, i.e., compacting the sample sprayed in a tube of the appropriate size by means of a metal rod. With reaction times of 8 and 40 ms, the authors' focus is to detect the reaction intermediate and not the rate of the reaction under study, which makes inhomogeneous sample packing less of a concern.

In the context of RFQ-HFEPR, Manzerova *et al.* [3] bring about an innovative method to freeze-quench reactants, reduce them to fine particles, and collect them. They report a method based on rotating copper wheels kept at a temperature of 80 K, on which the mixture of the reactants is sprayed through a home-built nozzle. The reactant mixture is thus freeze-quenched, and the sample is then "scraped" off the wheels and collected by tapping with a capillary suitable for 130 GHz EPR. Although such approach introduces the advantage of not having to handle static frozen particles floating in isopentane, the packing factor the authors report is limited to 0.5.

In the aforesaid methods a large amount of sample is required, which is often a disadvantage when working with biological samples. An improvement in this respect is introduced by Kaufmann *et al.* [4] and Pievo *et al.* [5], following a development by Cherepanov and de Vries [6] at low microwave frequency. They made use of micro-mixers, which require volumes of only hundreds of microliters. Kaufmann *et al.* also implemented a new way of collecting the RFQ samples, based on the idea of Manzerova *et al.* [3]. They use a rotating aluminum plate kept at a temperature of 80 K, on which the reagent mixture is sprayed and freeze-quenched. The packing efficiency is not mentioned, while in [5] a packing factor of 0.5 to 0.6 is reported.

From the cited literature thus emerges a serious difficulty in RFQ-HFEPR when it comes to collect RFQ particles and study them in a standardized, efficient, and reproducible way.

The present work originates from the premises cast by Nami in her PhD thesis [7], who explored the coupling of multi-frequency EPR with RFQ, based on an improved method [8] to pack and load capillaries for HF-EPR. Here we report the kinetic study of the binding reaction of sodium azide to myoglobin at 9.5 and 275 GHz. The correspondence of the results at both frequencies shows the feasibility of the approach at high microwave frequency. We demonstrate that EPR experiments at multiple microwave frequencies require only one series of RFQ samples.

## 2 Experimental

### 2.1 Materials

Equine-heart met-myoglobin, sodium azide ($NaN_3$), and DMSO were purchased from Sigma-Aldrich. $MnCl_2$ was purchased from Baker Chemicals. Myoglobin (Mb) was dissolved in sodium phosphate buffer 100 mM at pH 7.8, with the addition of 5% v/v DMSO and 50 µM of $MnCl_2$, to form a solution with concentration 2.4 mM. The concentration of the myoglobin solution was determined spectrophotometrically using the extinction coefficient $\varepsilon$ (505 nm) = 9.7 $mM^{-1}$ $cm^{-1}$.

Sodium azide was dissolved in phosphate buffer 100 mM at pH 7.8 to form a solution with concentration 24 mM. After RFQ mixing, the myoglobin and azide concentrations are 1.2 and 12 mM.

### 2.2 Sample preparation

Ten RFQ samples (named Mb1 to Mb10) were prepared with the RFQ apparatus (Update Instruments) and method described in [8] (2-mL syringes, ram velocity 3.2 cm $s^{-1}$, displacement 3 mm), at a mixing temperature of 21.5˚C, and at reaction times ranging between 2 and

**Table 1. Calculated reaction time of the RFQ samples.**

| RFQ sample | Mb1 | Mb2 | Mb3 | Mb4 | Mb5 | Mb6 | Mb7 | Mb8 | Mb9 | Mb10 |
|---|---|---|---|---|---|---|---|---|---|---|
| Calculated reaction time (ms) | 2.0 | 3.1 | 4.9 | 7.8 | 9.8 | 15.6 | 25.0 | 31.3 | 39.1 | 48.8 |

50 ms. These ten samples were initially measured at 9.5 GHz, and later the same were used for measurements at 275 GHz.

Table 1 summarizes the reaction times of the RFQ samples, which are calculated from the parameters used in the RFQ setup, and not corrected by the so-called dead-time (which includes the mixing time and the freezing time).

In addition, two corresponding samples were prepared, one without any sodium azide (labelled Mb0), meant to represent the sample before reaction, the other with sodium azide, meant to represent the sample after completion of the reaction.

## 2.3 Sample packing for 9.5 GHz EPR

The preparation of the RFQ samples for 9.5 GHz EPR in essence follows the procedure described by Nami [8]. Briefly, a 3 mm open quartz tube, equipped with a polypropylene filter at one end and connected to a syringe was submerged into the cold isopentane containing the minute particles from the RFQ. Using the syringe, the particles were sucked into the quartz tube resulting in the accumulation of the RFQ particles in front of the polypropylene filter while the isopentane passed through the filter. Typically, 4 to 5 cm of sample was collected after which the quartz tube was rapidly transferred into liquid nitrogen.

The packing procedure is described in detail in the SI. The quartz tubes prepared in this way were ready to be measured with the 9.5 GHz EPR spectrometer.

## 2.4 Sample packing and loading for 275 GHz EPR

The preparation of RFQ samples for 275 GHz EPR is more complicated than for 9.5 GHz EPR. The minuscule size of the capillaries used as sample holders (150 μm inner diameter) poses a problem. Accidental warming of the samples is easy and fast, in view of the tiny volumes involved. For this reason, they have to be handled at cryogenic temperatures.

A successful sample packing in capillaries for 275 GHz EPR requires dedicated tools, and is outlined in detail in the SI. The procedure, in essence reported in [7], is carried out in a polystyrene box half-filled with liquid nitrogen. Thanks to a flow of cold nitrogen gas blowing on the surface of the liquid nitrogen, the average temperature in the box within the first 10 cm from the liquid nitrogen surface is kept below -100˚C. In brief, the procedure includes the following steps:

- A RFQ sample contained in a quartz tube (a sample used for the EPR measurement at 9.5 GHz) is transferred from liquid nitrogen into dry ice for a few minutes to ensure the softening of the content. In this way, it is possible to collect the RFQ sample in the form of pellets by tapping the surface with a pre-cooled glass capillary. These pellets of sample are then dropped in a mortar filled with liquid nitrogen, to be ground and packed in a capillary for 275 GHz EPR.

- A customized quartz capillary of 150 μm inner diameter is used to collect the sample. The capillary is connected to a syringe, underpressure is made, the capillary is pre-cooled and dipped into the mortar, and the sample is sucked up.

- When at least 5 mm of the capillary have been filled with sample, the capillary is placed on a pre-cooled metal block (Fig 1C), whose function is that of keeping the capillary at low

temperature. The metal block is closed with its own lids, put in dry ice, and ready for loading.

As opposed to the straightforward sample loading for 9.5 GHz EPR, at 275 GHz special care and equipment is needed to ensure that the sample stays at cryogenic temperature during the whole loading procedure. To this end, at the moment of the sample loading, the spectrometer probe head has to be pre-cooled to a temperature between -90 and -80˚C. With the help of a loading stage that is clamped to the probe head (Fig 1B), the capillary is transferred from the metal block where it is accommodated (Fig 1C) into the probe head. The transfer of the capillary is done by lifting the block's lids one by one, starting with the lid farthest away from the probe head entrance, and pushing the capillary into the probe head. In this way the sample is never exposed to warm air and stays at cryogenic temperature from the beginning to the end of the loading procedure.

## 2.5 EPR measurements

RFQ samples are inherently heterogeneous because of both the quality of the mixing, and the amount and density of the packed particles in the collection tubes. In order to achieve quantitative results it is therefore pivotal to calibrate the EPR signals resulting from such samples. At 9.5 GHz, this is readily attained, because the low-field HS-Fe(III) and the high-field LS-Fe(III) are both detectable, and the total intensity of the heme-Fe(III) is distributed between these two forms [5]. At 275 GHz only the low-field HS-Fe(III) is detectable, and a reference signal is needed to normalize the Fe(III) signal. This is achieved by addition of $MnCl_2$ to the myoglobin solution [9], prior to the mixing in the RFQ apparatus. The $Mn^{2+}$ ion exhibits intense, sharp peaks around 9.845 T.

EPR measurements were performed with a 9.5 GHz (X-band) and a 275 GHz spectrometer. The former is an ELEXSYS E680 spectrometer from Bruker BioSpin GmbH, equipped with a He-flow ESR900 cryostat from Oxford Instruments. The latter is a home-built spectrometer [10], equipped with a He-flow CF935 cryostat from Oxford Instruments, and a home-built probe head with a single-mode cavity specifically designed for cw measurements [11]. The 275 GHz EPR spectrometer operates with a 14-Tesla superconducting magnet having an IPS120-10 power supply, both from Oxford Instruments.

The X-band spectra were recorded at a temperature of 20 K. Acquisition parameters: modulation frequency 100 kHz, modulation amplitude 0.5 mT, and microwave power 100 μW. Within a field range of less than 500 mT it is possible to detect both the HS-Fe(III) (low field), and the LS-Fe(III) (high field). The 275 GHz spectra were recorded averaging between 16 to 36 scans (depending on the sample) at a temperature of 10 K, and, given the high magnetic field required to operate at such high frequency, it is not convenient to record both the low-field HS-Fe(III) signal and the high-field $Mn^{2+}$ signal in one single spectrum. Acquisition parameters: modulation frequency 1.7 kHz, modulation amplitude 1.3 mT for Fe(III) and 0.3 mT for $Mn^{2+}$, microwave power 1.7 μW for Fe(III) and 0.8 μW for $Mn^{2+}$. Whenever shown, error bars represent the noise level of the averaged spectra.

## 3 Results and discussion

A common way of testing the performance of RFQ-EPR makes use of the binding reaction of sodium azide to myoglobin (Mb) [5] [12] [13]. At neutral pH, the heme iron of ferric myoglobin is in the high-spin (HS) S = 5/2 state. When an exogenous strong-binding ligand such as azide ($N_3^-$) replaces the axial water molecule, the Fe(III) converts its spin state to low-spin

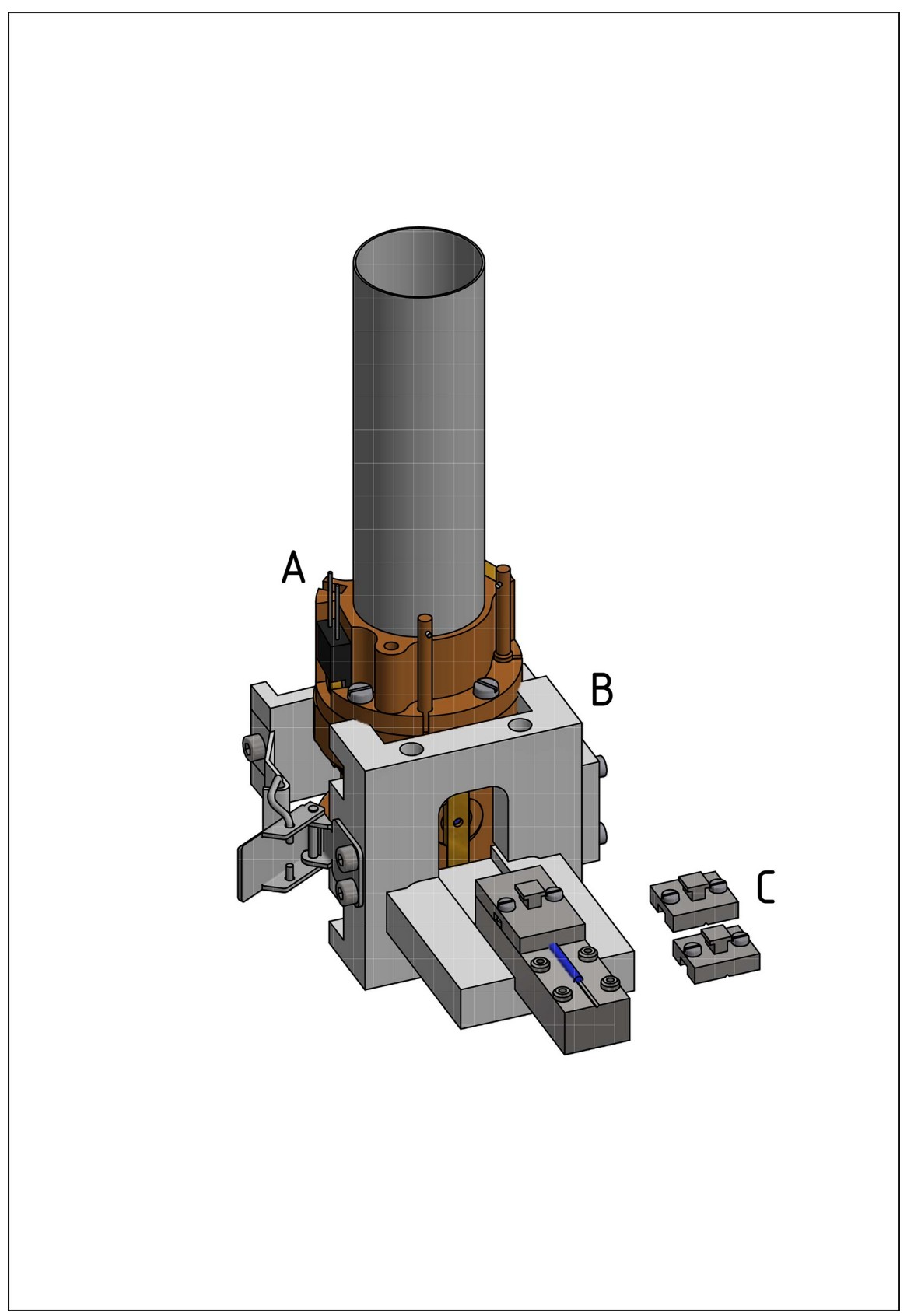

**Fig 1.** 3D rendering of the home-built assembly used for cold loading of RFQ sample tubes into the cavity of the 275 GHz spectrometer: (A), the probe head (brown) containing the 275 GHz cavity; (B), the assembly (grey) that is clamped onto the probe head with the sliding path for the metal block; (C), the metal block (dark grey), which holds the sample tube (blue), with the three lids to cover the sample tube in order to keep the sample cold.

(LS) S = 1/2. The spectral features of HS- and LS-Fe(III) are completely different, the former having an intense signal around g = 6, while the latter having a rhombic signal around g = 2.

The myoglobin-to-azide ratio of 1:10 in the present study allows to treat the binding reaction in terms of pseudo-first-order kinetics, so that the logarithmic ratio of the concentration of HS-Fe(III) at any reaction time t and at time t = 0 ($[HS]_t$ and $[HS]_0$, respectively) is proportional to the reaction time. The $k'$ is the apparent rate constant, and is the product of the actual rate constant $k$ and the azide concentration.

$$ln \frac{[HS]_t}{[HS]_0} = -k't \tag{1}$$

$$k' = k[N_3^-] \tag{2}$$

At 9.5 GHz, to derive the conversion of HS-Fe(III) to LS-Fe(III) from the EPR spectra, we made use of the HS-Fe(III) signal at times t, $(S_{HS})_t$, the LS-Fe(III) signal at times t, $(S_{LS})_t$, the HS-Fe(III) signal at t = 0 (i.e., reaction not yet begun), $(S_{HS})_0$, and the LS-Fe(III) signal at t = ∞ (i.e., reaction completed), $(S_{LS})_\infty$. According to Pievo *et al.* [5], the ratio of the HS-Fe(III) concentration at time t and at t = 0 can be expressed as

$$Y_t \equiv \frac{[HS]_t}{[HS]_0} = \frac{R_t}{R_t + \lambda} \tag{3}$$

in which

$$R_t = \frac{(S_{HS})_t}{(S_{LS})_t} \tag{4}$$

$$\lambda = \frac{(S_{HS})_0}{(S_{LS})_\infty}. \tag{5}$$

At 275 GHz, the conversion of HS-Fe(III) is derived from the HS-Fe(III) signal at times t after normalization by the $Mn^{2+}$ signal:

$$\left(S'_{HS}\right)_t \equiv \frac{(S_{HS})_t}{S_{Mn}} \tag{6}$$

Both at 9.5 GHz and at 275 GHz, the signals S were taken as the peak-to-peak amplitudes of the spectra at appropriate field values.

Fig 2 shows the X-band spectra of the myoglobin-azide RFQ samples Mb1 to Mb10, in the time range between 2.0 and 48.8 ms. A clear decay is seen of the low-field HS-Fe(III) signal at $B_0$ = 115.3 mT, accompanied by a proportional increase of the rhombic high-field LS-Fe(III) signal at $B_0$ = 241.9, 304.8, and 391.9 mT. Already for the Mb1 sample at t = 2.0 ms the LS-Fe (III) is detectable, while for the Mb10 sample at t = 48.8 ms the HS-Fe(III) signal has not completely disappeared, indicating that the reaction is not completed yet. Spectra of RFQ samples Mb5, Mb6, Mb8, and, to a lesser extent, Mb10, show a broadened signal in the range $B_0$ between 150 and 250 mT, clearly due to a contamination most likely during the preparation of the RFQ samples. In order to derive *Y* from the EPR spectra, the HS signal intensity was taken

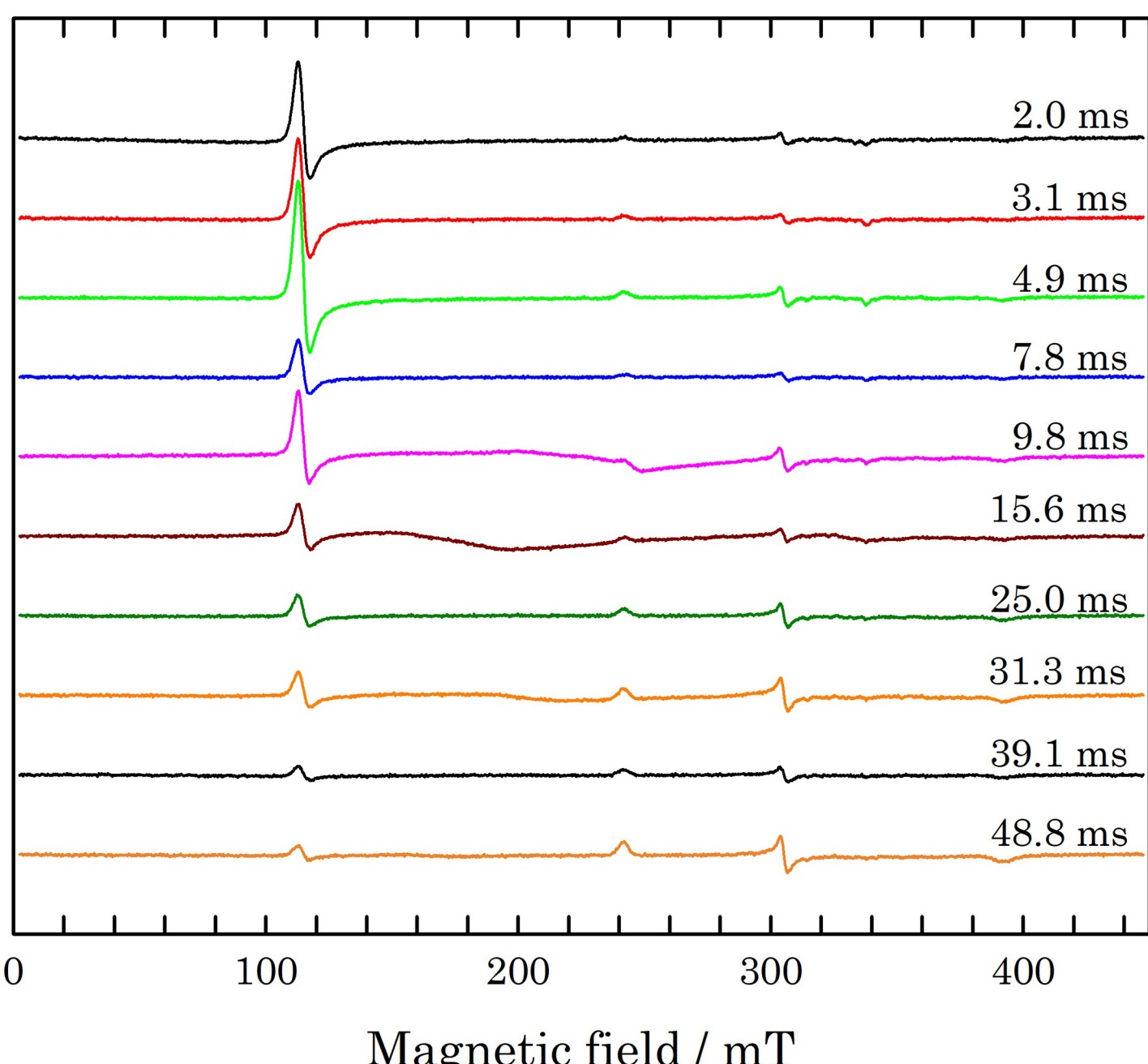

**Fig 2. Baseline-corrected, single scan 9.5 GHz cw EPR spectra of RFQ samples, Mb1 to Mb10 from top to bottom, in the time range between 2.0 and 48.8 ms.** The spectra were taken at 20 K.

at $B_0$ = 112.7 mT (max) and $B_0$ = 117.2 mT (min), while the LS signal intensity was taken from the central component of the rhombic spectrum at $B_0$ = 304.0 mT (max) and $B_0$ = 306.3 mT (min). The values of $Y$ are calculated according to Eq (3), with a value of 7.1 for $\lambda$. The latter was obtained from the spectrum of Mb0 and that of the completely reacted sample.

Fig 3A shows the decay of ln $Y$ versus the calculated reaction time. Through linear regression, the apparent rate constant $k' = 50 \pm 3$ s$^{-1}$ is extracted. By extrapolating the semi-

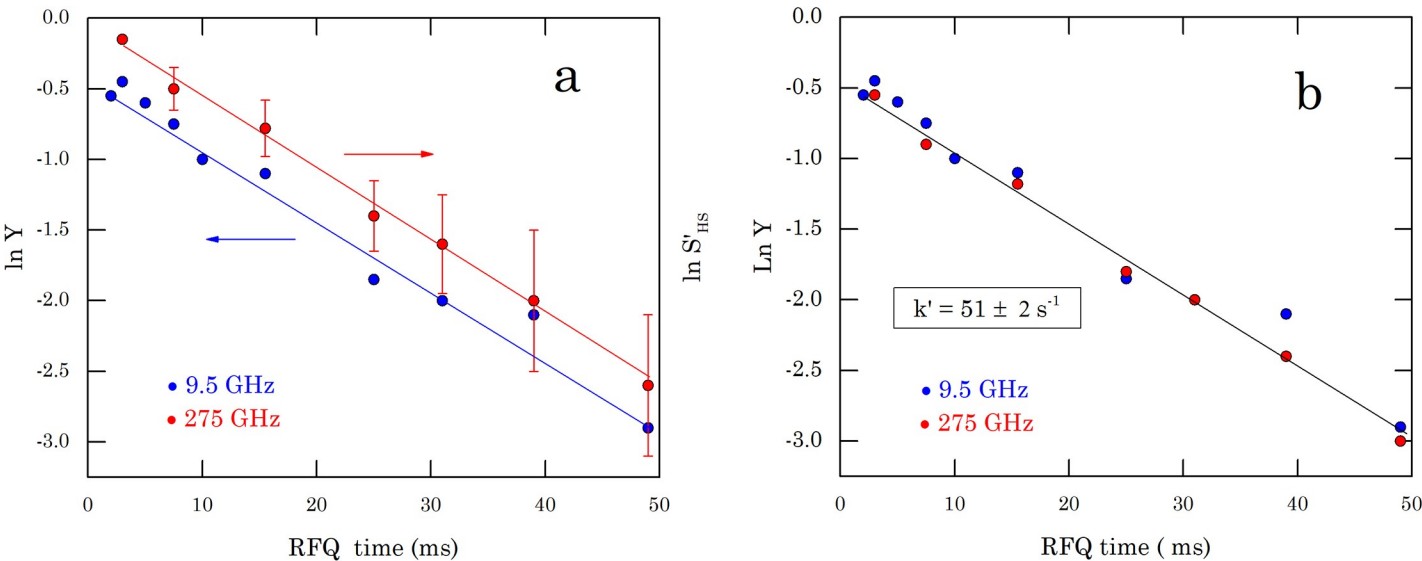

**Fig 3.** (a) $\ln Y$ at 9.5 GHz and $\ln S'_{HS}$ at 275 GHz as a function of the calculated reaction time, (b) Same as (a), but the data at 275 GHz shifted such that the intercept is zero at the dead time $t_d$.

logarithmic line to $\ln Y = 0$, a dead time $t_d$ of 7.9 ± 0.4 ms is obtained. The real reaction time differs by this amount from the calculated reaction time.

The 275 GHz spectra of the RFQ samples, normalized by the $Mn^{2+}$ signal, are given in Fig 4. In order to derive $S'_{HS}$ from these spectra according to Eq (6), the HS signal intensity was taken at $B_0 = 3.4780$ T (max) and $B_0 = 3.5948$ T (min), while the $Mn^{2+}$ intensity was taken from the sixth peak of the $Mn^{2+}$ spectrum at $B_0 = 9.8680$ T (max) and $B_0 = 9.8690$ T (min) (cf. inset of Fig 4). Fig 3A shows the decay of $\ln S'_{HS}$ versus the calculated reaction time. Through linear regression, the apparent rate constant $k' = 52 \pm 2$ s$^{-1}$ is extracted.

One series of samples is used for the EPR experiments at both microwave frequencies. Consequently, the dead time, determined by the RFQ process, obtained from the 9.5 GHz data applies to 275 GHz as well. This allows to combine the data from Fig 3A in one plot by shifting the 275 GHz data such that $\ln S'_{HS} = 0$ at the dead time of 7.9 ms. The result is shown in Fig 3B.

Indeed, the data at both frequencies now satisfy the same linear behaviour, which corresponds to an apparent constant $k' = 51 \pm 2$ s$^{-1}$. According to Eq (2), this value yields a rate constant $k = (4.2 \pm 0.2)\ 10^3$ M$^{-1}$ s$^{-1}$. This value of the rate constant lies in the range of reaction rates to be found in the literature for the myoglobin-azide reaction, under conditions similar to those of the present work [3] [5] [8] [12] [13] [14].

## 4 Conclusions

In conclusion, the correspondence of the results at 275 GHz with those at 9.5 GHz convincingly prove that the method described in the present work to couple RFQ to high-frequency EPR, is successful. The approach is based on the efficient and reproducible packing of RFQ samples in 150-μm sized capillaries suitable for 275 GHz EPR, and the safe and fast loading of the capillaries in the pre-cooled cryostat of our 275 GHz EPR spectrometer.

Besides this notable improvement over other methods reported in the literature so far, our approach also illustrates the use of only one single series of RFQ samples, to be used both at 9.5 and 275 GHz, and flexibly at any intermediate frequency. This advantage reduces the amount of material required for kinetic studies (which is particularly beneficial for biological samples),

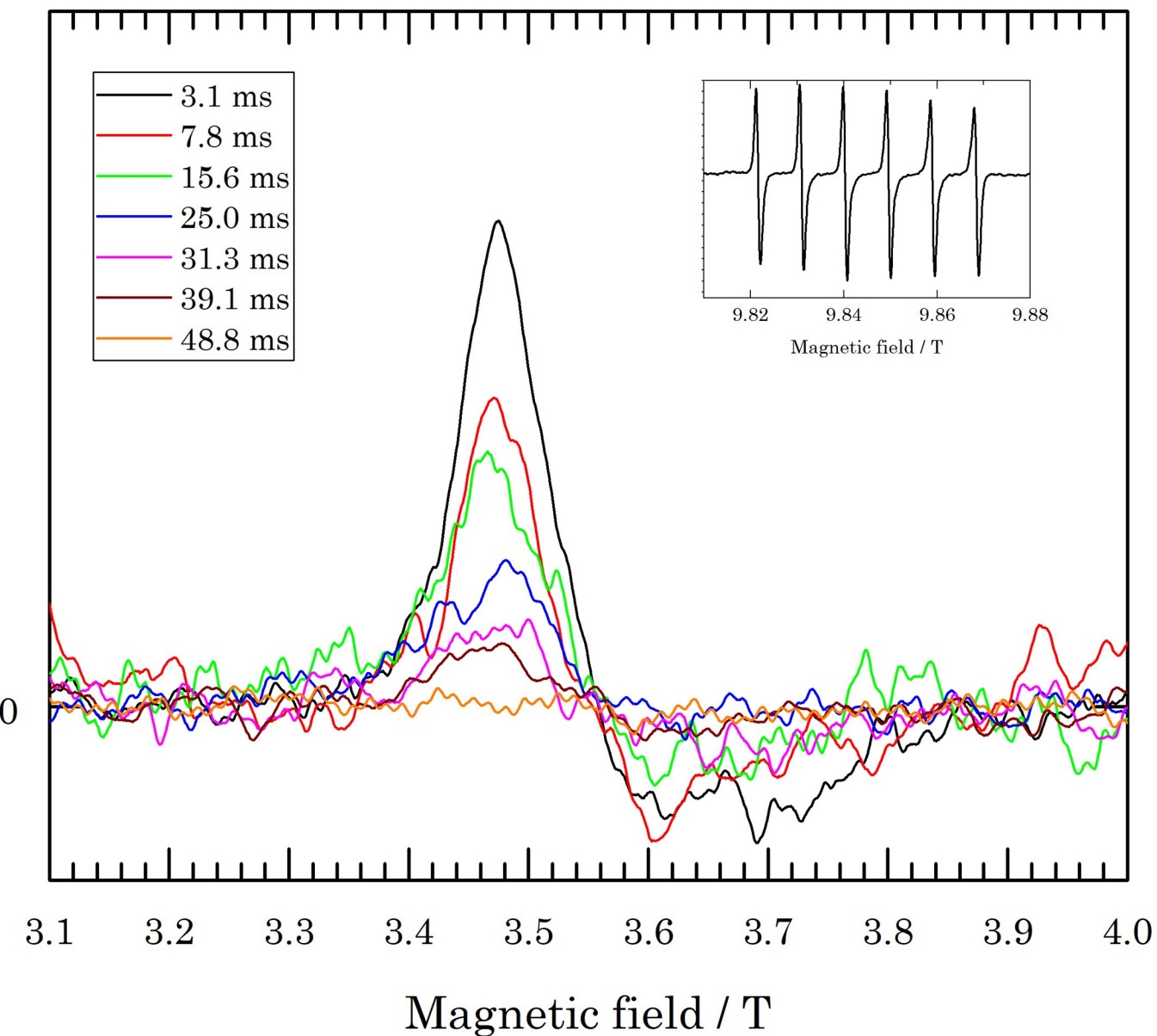

**Fig 4. Baseline corrected, Mn$^{2+}$-normalized, averaged 275 GHz cw EPR spectra of the RFQ samples.** The inset shows an example of an Mn$^{2+}$ spectrum. The spectra were taken at 10 K.

and improves the consistency of the method in that the same samples can be used for EPR investigations at different microwave frequencies.

## Supporting information

**S1 File. Detailed description of the sample packing at 9.5 GHz and at 275 GHz.** The data used for this article is publicly available on Figshare at the following link: 10.6084/m9.figshare.8982347.
(DOCX)

## Acknowledgments

The authors are thankful to Dr Mykhailo Azarkh and Dr Faezeh Nami for their contributions on the way to high-frequency EPR on RFQ samples.

## Author Contributions

**Conceptualization:** Edgar J J. Groenen.

**Formal analysis:** Edgar J J. Groenen.

**Investigation:** E. Gabriele Panarelli, Peter Gast.

**Methodology:** E. Gabriele Panarelli, Harmen van der Meer, Peter Gast.

**Supervision:** Edgar J J. Groenen.

**Writing – original draft:** E. Gabriele Panarelli, Edgar J J. Groenen.

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
