## [Decision Letter · Decision Letter 0]

6 Mar 2020

PONE-D-20-00403

Effective coupling of Rapid Freeze-Quench to High-Frequency Electron Paramagnetic Resonance

PLOS ONE

Dear Prof Groenen,

Thank you for submitting your manuscript to PLOS ONE. After careful consideration, we feel that it has merit but does not fully meet PLOS ONE’s publication criteria as it currently stands. Therefore, we invite you to submit a revised version of the manuscript that addresses the points raised during the review process.

We would appreciate receiving your revised manuscript by Apr 20 2020 11:59PM. To enhance the reproducibility of your results, we recommend that if applicable you deposit your laboratory protocols in protocols.io, where a protocol can be assigned its own identifier (DOI) such that it can be cited independently in the future. For instructions see: http://journals.plos.org/plosone/s/submission-guidelines#loc-laboratory-protocols

We look forward to receiving your revised manuscript.

Kind regards,

Francisco J. Blanco

Academic Editor

PLOS ONE

Journal Requirements:

Reviewers' comments:

Reviewer's Responses to Questions

**Comments to the Author**

1. Is the manuscript technically sound, and do the data support the conclusions?

Reviewer #1: Yes

Reviewer #2: Yes

2. Has the statistical analysis been performed appropriately and rigorously? 

Reviewer #1: Yes

Reviewer #2: N/A

3. Have the authors made all data underlying the findings in their manuscript fully available?

Reviewer #1: Yes

Reviewer #2: Yes

4. Is the manuscript presented in an intelligible fashion and written in standard English?

Reviewer #1: Yes

Reviewer #2: Yes

5. Review Comments to the Author

Reviewer #1: The paper contains innovative methodologies to study reaction with EPR spectroscopy and also technical development. The experimental data presented aim to the topics of the study.The manuscript is well written and clear.

Reviewer #2: General comments : The authors present an original manuscript about coupling preparation of samples using Rapid Freeze-Quench to High-Frequency Electron Paramagnetic Resonance.

Preparing samples for high-frequency EPR is often challenging specially in the case of biologic systems when one wants to obtain reactive intermediates and/or follow kinetics (and it is also true for lower EPR frequencies). Here, samples have to be made in very small capillaries, with the difficulties related to its possible rapid warm and then degradation of the intermediate. Here, the authors use a classical reaction, i.e. the binding reaction of azide to myoglobin, to demonstrate the feasibility of the method. Globally, the article is well written although the technical part of filling the samples for both EPR frequencies could me better explained, and could be illustrated by a scheme. The article is very interesting, and will be of great help for researchers interested in using high-field EPR for their studies. I still have some comments/questions to address to the authors (see attached file).

6. PLOS authors have the option to publish the peer review history of their article (what does this mean?). If published, this will include your full peer review and any attached files.

Reviewer #1: No

Reviewer #2: No

---

## [Author Response · Author response to Decision Letter 0]

15 Apr 2020

We have attached a file that contains specific responses to all comments by the reviewer.

---

## [Editor Report · Decision Letter 1]

17 Apr 2020

Effective coupling of Rapid Freeze-Quench to High-Frequency Electron Paramagnetic Resonance

PONE-D-20-00403R1

Dear Dr. Groenen,

We are pleased to inform you that your manuscript has been judged scientifically suitable for publication and will be formally accepted for publication once it complies with all outstanding technical requirements.

With kind regards,

Francisco J. Blanco

Academic Editor

PLOS ONE
---

## [Editor Report · Acceptance letter]

28 Apr 2020

PONE-D-20-00403R1 

Effective coupling of Rapid Freeze-Quench to High-Frequency Electron Paramagnetic Resonance 

Dear Dr. Groenen:

I am pleased to inform you that your manuscript has been deemed suitable for publication in PLOS ONE. Congratulations! Your manuscript is now with our production department. 

With kind regards,

on behalf of

Dr. Francisco J. Blanco 

Academic Editor

PLOS ONE